# Estimation of Emissions at Signalized Intersections Using an Improved MOVES Model with GPS Data

**DOI:** 10.3390/ijerph16193647

**Published:** 2019-09-28

**Authors:** Ciyun Lin, Xiangyu Zhou, Dayong Wu, Bowen Gong

**Affiliations:** 1Department of Traffic Information and Control Engineering, Jilin University, Changchun 130022, China; linciyun@jlu.edu.cn (C.L.); zhouxy17@mails.jlu.edu.cn (X.Z.); 2Jilin Engineering Research Center for ITS, Changchun 130022, China; 3Department of Civil, Environmental and Construction Engineering, Texas Tech University, Lubbock, TX 79409, USA; jasond.wu@ttu.edu

**Keywords:** traffic pollution, emissions, GPS data, MOVES

## Abstract

Emissions from the transport sector are responsible for a large proportion of urban air pollution. Scientific and efficient measurements on traffic pollution emissions have already been a vital concern of decision makers in environmental protection. In China or other counties, many high-technology companies, such as Baidu, DiDi, have a large number of real-time GPS traffic data, but such data have not been fully exploited, especially in purpose of estimation of vehicle fuel consumption and emissions. In this paper, the traditional MOVES (Motor Vehicle Emission Simulator) model has been improved by adding the real-time GPS data and tested in representative signalized intersection in Changchun, China. The results showed that adding the GPS data sets in the MOVES model can effectively improve the estimation accuracy of traffic emissions and provide a strong scientific basis for environmental decision-making, planning and management.

## 1. Introduction

The rapid development of urbanization has brought a lot of air pollution problems. The haze weather has seriously affected the health of urban residents. Some researchers have shown that traffic emissions have become an important constitute of pollution emissions. Combining urban traffic, pollution with air quality, the air pollution level in various regions of China is directly displayed [1]. With the vigorous development of transportation industry, carbon dioxide caused by excessive use of multiple energy sources in transportation industry has aggravated the greenhouse effect of the earth [2,3]. For the numerical statistics of traffic pollution emission, emission model is relatively multitudinous. In the beginning of emission model research, emission model was built on VSP (Vehicle Specific Power) based on emission rates, and that made it flexible to apply most places in the world as long as users have VSP related operating mode distributions [4]. With the increasingly rich functions of smartphones, VSP model is combined with the smartphone’s location information to predict the vehicle fuel consumption in real time [5]. Gradually, some researchers analyzed the relationship between the behavior of drivers and vehicle fuel consumption emissions through driver’s smartphone positioning function [6,7]; moreover, distinguishing different driving behavior could accurately gather statistics of the car’s fuel consumption emission [8,9]. With the in-depth study of emission models, different emission standard models appeared in different countries. The IVE (International Vehicle Emissions) model was set up specifically for developing countries and it was more suitable for macroscopic traffic analysis, assessing the total amount of combined emissions in some areas [10,11]. The continuous improvement of the IVE model was used in evaluating the impact of different road types on vehicle emissions. Meanwhile, different design types of intersections were considered. The roundabout, a kind of special intersections was analyzed to contrast with ordinary intersection emissions [12,13]. Complex traffic scene and application joined in emission model [14] and constructed emission model used the average speed of the vehicle, like the MOBILE models and EMFAC models. Meanwhile, MOVES emission model released by the environmental protection agency of USA is operable and widely used [15,16].There are many parameters in the complex model, but the actual emission operation results are not significantly improved compared with the previous model [17].

In the area of traffic signal control, there was a relationship between the signalized intersection and the exhaust gas discharged. Vehicular portable measurement system was used to analyze roundabout intersection and traffic signal intersection instantaneous emissions of CO_2_, NO_x_, and CO [18,19]. Signal algorithm optimization showed obvious delay reductions compared to initial control algorithm [20,21]. With the same cycle length and platoon ratio, the time when the vehicle arrives at the intersection still had a certain degree of fluctuation in emission [22]. Typically, the assessment of vehicle emissions was mainly divided into two parts: one was the macroscopic study of the road network. Model considered the impact of traffic flow on the road environment [23,24]. Another microscopic study of the road network was to analyze the different motion states of the vehicle [25]. The improvement of microscopic traffic parameters in urban roads got multifarious fluctuation in signal-controlled road traffic flow [26]. Some model made systematic progress through improving the data assimilation methods and related models [27,28,29,30]. 

In the previous research of emission models, most of them used historical data. The use of real-time data is still rare. Furthermore, many Chinese Internet companies (e.g., DiDi, Tencent, and Baidu) collect a large amount of real-time vehicle information. However, these real-time traffic data are not used in traffic emission studies. Real-time traffic data can bring effective analysis to short-term urban traffic situation. Especially in haze and other polluting weather, using vehicle GPS data can quantify regional exhaust emission and provide effective emission data to relevant environmental protection parts in time. Considering the problems mentioned above, real-time GPS data is added to the flexible MOVES model in this paper to further carry out the data-driven emission research.

## 2. Materials and Methods 

### 2.1. Data Sources

The data comes from the GPS data of the vehicles in Changchun collected by DiDi Internet Company. DiDi is a technology company focused on transportation. It is worth mentioning that Didi establishes a flexible travel mode of users by the internet. The vehicle data of the network records the driving data of multiple vehicles in one day. Test data uses the evening rush hour from 5 p.m. to 6 p.m. The GPS data contains 86 intersections. The types of vehicle have Volkswagen Magotan, FAW Jetta, FAW Pentium and Toyota Corolla. The actual test data included 562 Volkswagen Magotan vehicles, 1335 FAW Jetta vehicles, 723 FAW Pentium vehicles, and 289 Toyota Corolla vehicles. The speed range of the vehicles is less than 100 km/h. The test drivers of the vehicles were in normal driving state. Figure 1 is the distribution map of the intersections from Google map. The studied intersection is Ziyou road and Tongzhi Street. Figure 2 is an aerial map of the study intersection.

The GPS data have an approximate sampling interval of 10 s. In practice, as to obtain higher location accuracy to provide high-precision navigation in intersection, the interval of GPS data is shorter when vehicles entering the intersection. Different time interval patterns are selected based on road conditions and road section length. The actual return transmission interval is set at about 1 s to ensure the accuracy of the intersection emission test. The interval is not set at 1 s on the normal driving road. In fact, considering that the shorter the time needs, the greater the power of the equipment and the more prone to failure of the equipment operation. Subsequent research uses 1 s as the interval for validation. A description of the data is shown in Table 1, where “Date” means records generating, “ID” represents the vehicle’s ID, “Longitude” and “Latitude” are the geographic location of the vehicle, “velocity” represents different ID vehicle’s velocity, “Directional angle” was recorded vehicle’s steering angle. In order to estimate emissions more accurately, different temperatures are needed, which are −3 °C and −2 °C on 14–15 February, respectively. The actual vehicle emission testing instrument adopted in this study is the OEM-2100 produced by CATI company in the United States. The instrument calculates the exhaust volume flow rate according to the measured engine parameter data, and calculates the instantaneous mass emission rate based on the measured exhaust volume percentage.

### 2.2. Data Processing

Some of the vehicles are affected by the intersection signal control. When the vehicle speed is zero, the vehicle will be in an idle state. Then, the acceleration is passed and finally the vehicle enters the working condition at a constant speed. The speed of the networked vehicles is not the same. Different speeds correspond to different driving conditions. It is very important to divide the running state of the vehicles at different speeds. The vehicle speed is divided into working conditions, and idle condition. The overall flow chart for the improved model is shown in Figure 3 as follows.

The classification accuracy of KNN (k-Nearest Neighbor) algorithm is relatively high, according to the multi-category classification of vehicle running state. The operation process of data classification is convenient and the classification applicability of the algorithm is high, so the real-time data can be used for fast classification of vehicle motion states. The processing of the KNN algorithm is as follows:(1)Calculate the distance between the test data and each training data;(2)Sort according to the increasing relationship of distances;(3)Select the K points with the smallest distance;(4)Determine the frequency of occurrence of the category of the first K points;(5)Return to the category with the highest frequency among the top K points as the prediction classification of the test data.

In KNN, by calculating the distance between objects as a non-similarity index between objects, the matching problem between objects is avoided, where the distance is generally Euclidean distance.
(1)$$d\left(x,y\right)=\sqrt{\overset{n}{\underset{k=1}{\sum }}{\left(x_{k}-y_{k}\right)}^{2}}$$

The k cluster centers of the sample are u_1_, u_2_,…, u_k_, the number of cluster samples is O_1_, O_2_,.…,O_k_, and the scoring error of the sample is the error function J(x):(2)$$J(u_{1},u_{2}..u_{k})=\;\frac{\sum_{j=1}^{k}\sum_{i=1}^{O_{j}}{\left(x_{i}-u_{j}\right)}^{2}}{2}$$

The error function is derived to take the small value as the optimal solution of the function at this time, which is related to the driving conditions of acceleration, deceleration, idle speed, and uniform speed during the running of the vehicle. The speed is generally zero under idle conditions. The speed range is divided into three categories, which are the final targets. The speed can be divided into three clusters (the number of center points of k is 3). The normal vehicle speed will maintain a certain steady speed, and the uniform driving situation will be divided into a cluster. When the vehicle is acceleration and deceleration during entering and departing the intersection, the classification standard can determine the acceleration and deceleration of the vehicle by using the positive or negative of the speed change. Quickly distinguish vehicle speeds to identify vehicle operating conditions and further accurately calculate vehicle exhaust emissions.

The speed of NO.81457 vehicle is processed at random during the early rush hours. The original data of vehicle speed are shown in Figure 4. The figure’s horizontal axis represents the speed of the vehicle and the vertical axis represents the rescaled distance cluster combination. The unit of velocity is kilometers per hour in Figure 4.

After improved KNN clustering, the results are shown in Figure 5. The figure’s horizontal axis and vertical represents rescaled distance cluster combination.

The vehicle speed judgment point distinguishes between different stages, acceleration, uniform speed, and deceleration. The three vehicle operating conditions correspond to Figure 3.

The blue point indicates that the vehicle speed is at a constant speed condition, the red point indicates that the vehicle speed is under the accelerated driving condition, and the green point indicates that the vehicle speed is under the deceleration condition.

### 2.3. Improved MOVES Emission Model Based on GPS Data

#### 2.3.1. Combined VSP Emission Model

Vehicle speed in GPS data is relatively accurate and has a high real-time performance, which can quickly evaluate vehicle emission at intersections from the perspective of data. According to the analysis of vehicle dynamics, the exhaust emissions of vehicles have the most direct relationship with the engine output of the vehicle. The output power variation factor of the engine has an important relationship with the instantaneous speed of the vehicle. The specific power model can be used to correlate vehicle emissions with vehicle operating conditions and to accurately quantify vehicle exhaust emissions. The speed in the GPS data is the original value, so the error of the research using the speed is small. The VSP calculation equation [25] for light vehicles is as follows:(3)$$P_{\mathrm{VSP}}=v\cdot \left(1.1\cdot a+9.8at+0.132\right)+0.000302\cdot v^{3}$$
where: $P_{\mathrm{VSP}}$ is the Vehicle Specific Power; v is the vehicle instantaneous speed; a is the vehicle acceleration; t is the time.

It can be concluded from the above formula that the vehicle will produce different values under different motion states. Under the deceleration condition of the vehicle, the specific power is negative. Fitting analysis of vehicle emissions and vehicle transient conditions, domestic and foreign scholars carry out a variety of studies. The specific power model will be divided into quantitative intervals of vehicle conditions, clustering analysis can be divided into a number of intervals. Table 1 shows the exhaust gas emission rates of CO, CO_2_, NO_x_, HC, etc. This range test vehicle is a light power vehicle with an engine displacement of less than 3.5 L and a mileage of 105 km. 

The test acquires the data as the instantaneous speed of the vehicle, and the specific power formula coefficient is changed as follows:(4)$$P_{\mathrm{VSP}}=v\cdot \left(1.1\cdot v/t+9.8v+0.132\right)+0.000302\cdot v^{3}$$

Through Table 2 [25], it can calculate the different exhaust emission unit indexes at different speeds during the driving process of the vehicle. During the test, the floating vehicle data can be exchanged by using the Internet. The various indicators of the vehicle’s motion state can be monitored in real time, and the above specific power calculation formula is further used. The real-time test vehicle speed data is inserted to obtain real-time monitoring and acquisition of vehicle exhaust emissions in real time.

#### 2.3.2. Improved MOVE Model Based on GPS Data

Through the above-mentioned specific power exhaust emission rate table, it can be known that the exhaust gas of the vehicle will change from time to time. Furthermore, the amount of exhaust gas discharged under different working conditions during the running of the vehicle changes accordingly. There are four working conditions during the running of the vehicle, accelerated working condition, uniform working condition, decelerated condition and idle working condition. The deceleration driving includes the special vehicle condition of neutral sliding. Since the speed of the neutral sliding is not necessarily zero, the exhaust emission of the vehicle in the neutral sliding can be put into the deceleration driving condition to calculate the exhaust emission of the vehicle. Vehicle speed information is extracted from GPS data at signalized intersections. Although different temperatures could have little impact on emission, the current car fuel tank has temperature protection and the exhaust emission temperature is higher than temperature. The GPS data is used in simulation without separated analysis. Vehicle emissions can be divided into four parts from different vehicle conditions. The total amount of exhaust emissions during vehicle operation can be given by the following formula:(5)$$E_{\mathrm{all}}=E_{\mathrm{ac}}+E_{\mathrm{sl}}+E_{\mathrm{un}}+E_{d}$$
where: $E_{\mathrm{all}}$ is the total emission; $E_{\mathrm{ac}}$ is the accelerated emission; $E_{\mathrm{sl}}$ is the deceleration emission; $E_{\mathrm{un}}$ is the uniform discharge; $E_{d}$ is the idle emission.

Nevertheless, every vehicle speed can be quickly classified from the data. But the macro vehicle queue does not reflect precise emissions. Different specific power interval was corresponding to different speed section from Table 2. Vehicles would not cause idle state when vehicles pass through the intersection in the green light time. In this state, vehicle queue running state is relatively simple. Using vehicle traffic speed and time could analyze vehicle queue’s exhaust emission. But vehicles in red time couldn’t pass through the intersection. The vehicle status in the vehicle queue was correlated with each other. The macro method was used to analyze vehicle queue and then the state of each car was analyzed from the four microstates to combine with VSP model. On the whole, the process of driving to a signal-controlled intersection could be divided into the following two cases.

(a) Vehicle passing through the intersection

In such cases, the speed of the vehicle at the intersection will generally change, but the speed of the vehicle will not change too much. The vehicle is not in an idle state and the total amount of vehicle emissions E is as follows:(6)$$E=\sum_{i=1}^{N}\sum_{j=1}^{M}e_{j}$$
where: N is the total number of traffic through the intersection; $e_{j}$ is each vehicle corresponds to the driving conditions, and the speed corresponding to the speed in different specific power intervals; M is the vehicle travel time.

(b) The vehicle has not passed the intersection

When the vehicle passes through the signal-controlled intersection, the vehicle’s driving condition will undergo a series of changes. Further, the vehicle motion state is divided into a uniform driving condition, an accelerated driving condition, an idle working condition, and a deceleration working condition. Using kinematics to analyze vehicle motion, the vehicle travel distance S is as follows:(7)$$S=V_{0}t+\frac{{\mathrm{at}}^{2}}{2}$$
where: $V_{0}$ is the initial speed of the vehicle.

The density of traffic on the road network can be expressed in a generalized velocity density model:(8)$$V=V_{f}{\left(1-\frac{K}{K_{j}}\right)}^{n}$$
where: V is the speed of Internet-connected vehicles; $V_{f}$ is the maximum traffic speed; n is the greater than zero real number; $K_{j}$ is the maximum traffic volume density

In conclusion, the relationship between the number of vehicles on a road and the speed can be derived:(9)$$k=k_{j}\left[1-\left(1-{\left(\frac{V}{V_{f}}\right)}^{\frac{1}{n}}\right)\right]$$
(10)$$N=k*l=k_{j}\left[1-\left(1-{\left(\frac{V}{V_{f}}\right)}^{\frac{1}{n}}\right)\right]l$$
where: N is the number of vehicles in each section; l is the length of each section.

Number of vehicles on each entrance $N_{i}$:(11)$$N_{i}=k_{j}\left[1-\left(1-{\left(\frac{V}{V_{f}}\right)}^{\frac{1}{n}}\right)\right]l_{i}$$

The following is analysis of the networked vehicles in the following four conditions.

(1) Vehicle deceleration condition

When the vehicle is in the deceleration condition, the acceleration of the vehicle is less than zero during the driving process. Then the vehicle trajectory of the individual vehicle is simplified and analyzed. The vehicle enters the intersection to perform deceleration, where in the vehicle has deceleration, and there may be multiple vehicles decelerating to parking multiple times. The situation of the line is considered comprehensively. Single vehicle exhaust emission E_1_:(12)$$E_{1}=a\sum_{j=1}^{M}e_{j1}$$
where: a is the single vehicle parking times.

Total emissions from traffic on the road segment E_11_:(13)$$E_{11}={\mathrm{NE}}_{1}$$

(2) Vehicle idle condition

The vehicle enters the signal control intersection. When the red light is on, the vehicle waits for the traffic at the stop line. At this time, the vehicle speed is zero and the acceleration is zero. The vehicle emissions can be obtained by using the [0,1) interval value in the power emission rate table. E_2_ is emission of vehicle idling conditions.
(14)$$E_{2}=a\sum_{d=1}^{M}e_{j2}$$

(3) Vehicle acceleration condition

The vehicle waits for the end of the red light and starts to accelerate out of the vehicle stop line. The kinematic angle analysis can understand the acceleration condition as the reverse direction deceleration condition. The exhaust emission amount E_3_ of the vehicle flow is Equation (14).
(15)$$E_{3}=\mathrm{Na}\sum_{j=1}^{M}e_{j3}$$

(4) Vehicle uniform speed condition

The changes of vehicle driving conditions on specific road sections are more complicated. Many algorithms for vehicle emissions are not accurate for the analysis of the state of uniform driving conditions of vehicles. This paper uses real-time Internet information transmission technology to further accurately obtain the state of vehicle running. A reasonable speed interval divides the various driving conditions of the vehicle, so that the data can clearly analyze various working conditions of the vehicle. Under the uniform working condition of the vehicle, the vehicle emission can be calculated by using the differentiated speed under constant driving conditions to reduce the interference of the speed of the other working conditions on the discharge of the uniform working condition. At this time, the vehicle exhaust emission is:(16)$$E_{4}=\mathrm{Na}\overset{T}{\underset{t=1}{\sum }}\overset{M}{\underset{d=1}{\sum }}e_{j4}$$

In summary, vehicles emit different emissions under different driving conditions. In order to simplify the calculation of various emission vehicles on the road section, the vehicle analysis of road vehicles is superimposed. Then the emissions of imported vehicles from the north–south direction and the east–west direction are uniformly added. It is concluded that there is a signal to control the total amount of vehicle emissions at the intersection, and the total amount of single-cycle vehicle emissions after the single-segment is simplified:(17)$$E_{\mathrm{all}}=\overset{N_{i}}{\underset{i=1}{\sum }}\overset{a_{i}}{\underset{i=1}{\sum }}\left(\overset{M}{\underset{j=1}{\sum }}e_{j1}+\overset{M}{\underset{j=1}{\sum }}e_{j2}+\overset{M}{\underset{j=1}{\sum }}e_{j3}+\overset{T}{\underset{t=1}{\sum }}\overset{M}{\underset{d=1}{\sum }}e_{j4}\right)$$
where: $N_{i}$ is the number of vehicles on each entrance.

## 3. Results

MOVES model, actual measured emission data and GPS-based improved model are compared in this paper. The road section is a standard two-way road intersection as an example. Firstly, the micro-simulation model is used to calculate the vehicle exhaust gas, and then the measured data were compared, and the final simulation analysis confirmed the effectiveness of the proposed emission method.

### 3.1. Road Condition Analysis

The test intersection is controlled by four phases (including left-turn phase) at Changchun Freeway and Tongzhi Street Interchange. The signal period is 132 s, and the speed of the given vehicle section is 40 km/h. Assumed that the vehicle acceleration takes 2 m/s^2^. The saturation flow rate of each straight lane is 1800 pcu/h, and the saturation flow of 1500 pcu/h in the left lane. The data is used in each direction 150 m away from the intersection, the lane width is 3.5 m, and the vehicles are tracked within 150 m of the intersection. The total parking delay time is 31.3 h, and the total number of parking times is 2342. The GPS data is transmitted from time to time using the Internet. The evening peak period between 5:00 and 6:00 was selected. Combining Synchro8 road network control simulation software with emission simulation software MOVES, the simulation results show the emission of intersection with signal control. In simulation, traffic flow rate and signal timing data are consistent with the field of the traffic network and road condition. The running time is setting as 3600 s, and setting the warm-up period is 0 to 10 minutes so as to reduces the fluctuation of running parameters. The actual channelization diagram of the intersection is shown in Figure 6. 

The actual signal timing diagram of the test intersection is shown in Figure 7.

Select the test intersection traffic flow chart as shown in Figure 8.

### 3.2. Model Checking

In the above, the GPS-based improved model is constructed with the specific power. 2909 vehicles in the peak hour are selected for emission assessment. The emission research mainly focus on CO_2_ and NO_x_. In the parameters, the number of vehicle stops is 2342. The delay time is 31.3 hours. The accumulated statistical duration is one hour. The exhaust gas emission statistics are calculated by using the networked vehicle exhaust gas meter, and the model is mainly for the public. Magotan, FAW Jetta, FAW Pentium, Toyota Corolla. The average specific power of the four models is shown in Table 3 below.

The specific power interval is selected as the [39, +∞) interval in Table 1, and the specific power-related exhaust emissions are shown in Table 4. 

The actual test data counted 562 Volkswagen Magotan, 1335 FAW Jetta, 723 FAW Pentium, and 289 Toyota Corolla. Combined with Table 4, we use the statistical delay time to calculate the exhaust emission of vehicles in idling state. After classifying the remaining speed, the exhaust emission statistics are carried out according to VSP formula to calculate the emissions of vehicles in different working conditions. The total emissions can be obtained by summation of each state. In the actual measured data, Volkswagen Magotan’s CO_2_ emissions are 27,928.03 g, NO_x_ emissions are 256.17 g, FAW Jetta’s CO_2_ emissions are 66,325.63 g, NO_x_ emissions are 608.53 g, FAW Pentium’s CO_2_ emissions are 35,922.34 g, NO_x_ emissions are 329.56 g, Toyota Corolla The CO_2_ emission is 14,358.00 g, and the NO_x_ emission is 131.74 g. The information on the exhaust emissions of the network is as shown in Table 5.

### 3.3. Simulation Analysis

In the simulation experiment, the traditional MOVE emission model and the improved emission model combined with GPS data were compared and analyzed the various parameters of the MOBILE emission software are set as follows:
The displacement of the engine is designed in the above four models. The Volkswagen Magotan and Toyota Corolla are set to 1.8 L, and the FAW Jetta and FAW Pentium are set to 1.5 L.The load and self-weight setting of the vehicle, Volkswagen Magotan is 1285 kg, FAW Jetta is 1110 kg, FAW Pentium 1285 kg, Toyota Corolla is 1265 kg.Environmental parameters and maintenance status settings. The weather is good and the simulation settings are consistent during the actual test. The vehicles are in good condition.

Combined with MOBILE emission software, the vehicle specific information is input. The settings are divided into four types of vehicles for setting), and the simulated emission indicators are obtained as CO_2_ and NO_x_ emissions. The exhaust emissions obtained by the three methods are shown in Table 6.

## 4. Discussion

In this paper, comparative analysis shows that the improved model based on GPS data most closely with the actual measurement, construct the emission model emissions of CO_2_ emissions and the actual error is 3.84%, NO_x_ emissions and the actual error is 5%, combined with the traditional MOVE model error is 6%. It is concluded that CO_2_ emissions and the actual emissions of NO_x_ emissions and emissions actual error was 9.12%.

In this paper, through joining real-time GPS data analyzes the vehicles entering the intersection emissions. Compared with the traditional MOVES model, the traditional model studies vehicle emissions by using variables such as vehicle model environment and speed. Moves model does not consider the change of vehicle conditions in the changing process of signalized intersections and the necessary delay caused by the signal control at the intersection. The real-time GPS data added by the improved model can effectively improve the accuracy of emission quantification, and analyze emissions by combining the timing of different signal intersections. Combined with the process of vehicles entering the intersection, it can reasonably analyze the emission of vehicles under different conditions. The exhaust emission obtained by the three methods can be obtained. The error between the emission calculated based on the GPS improved model and the actual measured value is significantly smaller than that obtained by the traditional MOVES model. This further verifies that the GPS-based improved model is better. Adding real-time GPS data to the model can improve the accuracy of exhaust emission and related parameters in macroscopic and microscopic level. It is significant to improve the efficiency of GPS data processing, which is an important direction in the future.

## 5. Conclusions

Based on the real-time GPS data from high-technology companies, we have greatly improved the data-driven emission research at intersections. We have found that integrating traditional emission model with real-time data can monitoring the emissions of intersection in real-time which can be used as an objective in signal optimization for signalized intersection, as to reduce the pollution of traffic emissions. In addition, environmental protection agencies can quickly make corresponding policy and air monitoring through quantified traffic exhaust emissions, which can reduce the pollution to a certain extent.

Further study is needed for different driving conditions, and differences driving behaviors need to be further studied in the follow-up study. MOVES emission model analyzes speed and other environmental factors on the road to emission estimates. However, processing vehicles especially in a signalized intersection into the queue status hasn’t been able to study effectively. In this study, we differentiate between queuing vehicles with queuing model, but the accuracy of the queuing model needs to be improved in the following study. As we all know, the more accuracy to identify the queuing status of the vehicles in the intersection, the more accuracy to obtain the emissions of the vehicles. The real-time GPS data can further improve the accuracy of the data in the MOVES model and better evaluate the exhaust emission at the signalized intersection.

## Figures and Tables

**Figure 1 ijerph-16-03647-f001:**
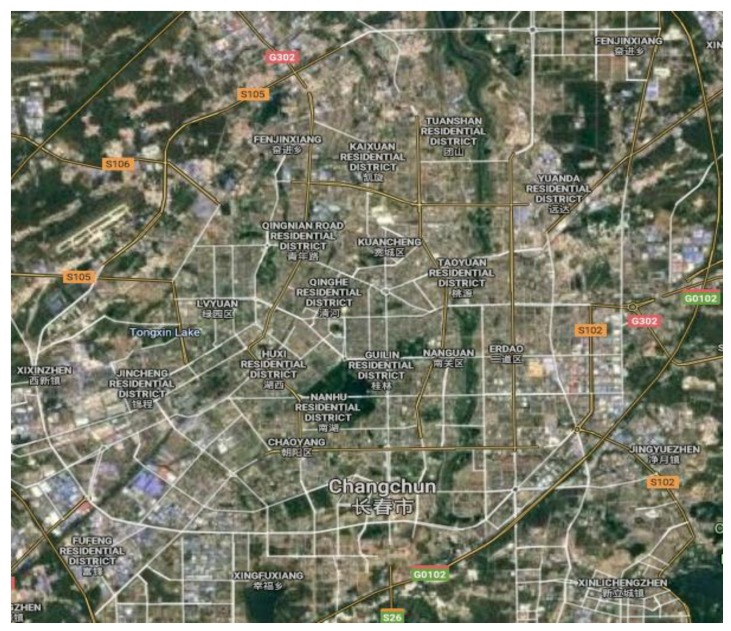
The intersection distribution map in GPS data.

**Figure 2 ijerph-16-03647-f002:**
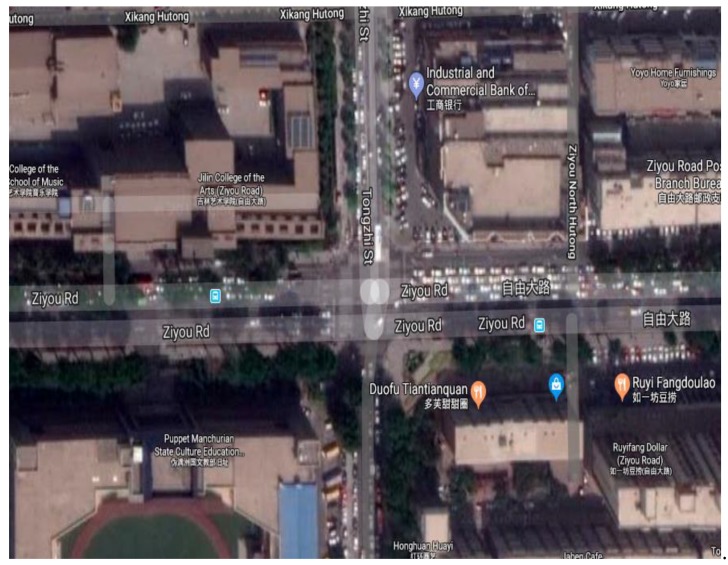
The aerial view of the study intersection.

**Figure 3 ijerph-16-03647-f003:**
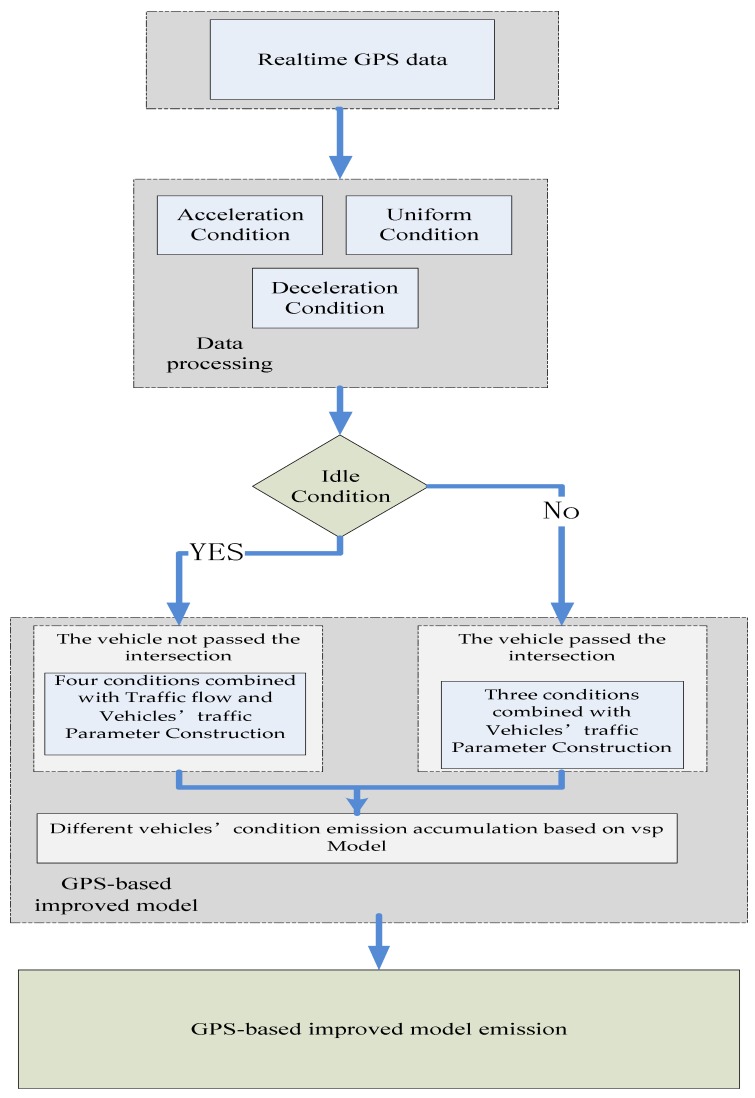
GPS-based improved model’s flow chart.

**Figure 4 ijerph-16-03647-f004:**
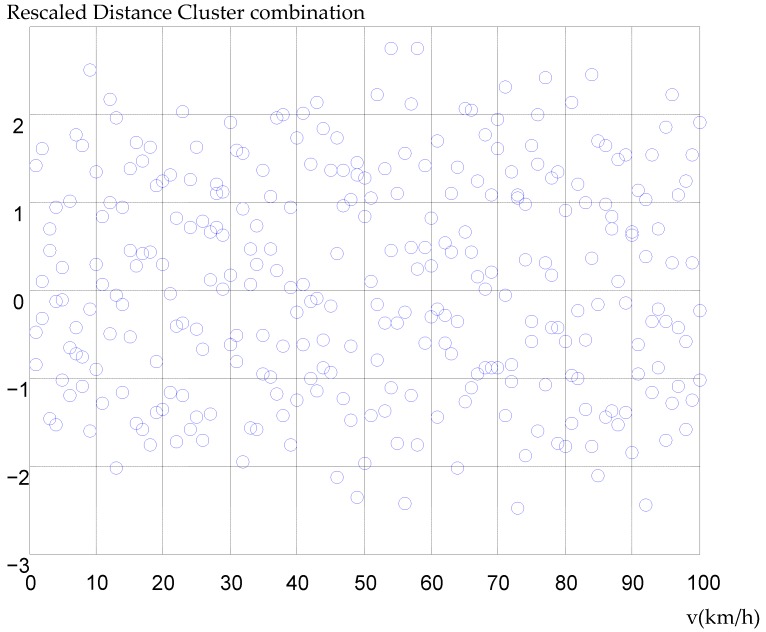
The original data of vehicle speed.

**Figure 5 ijerph-16-03647-f005:**
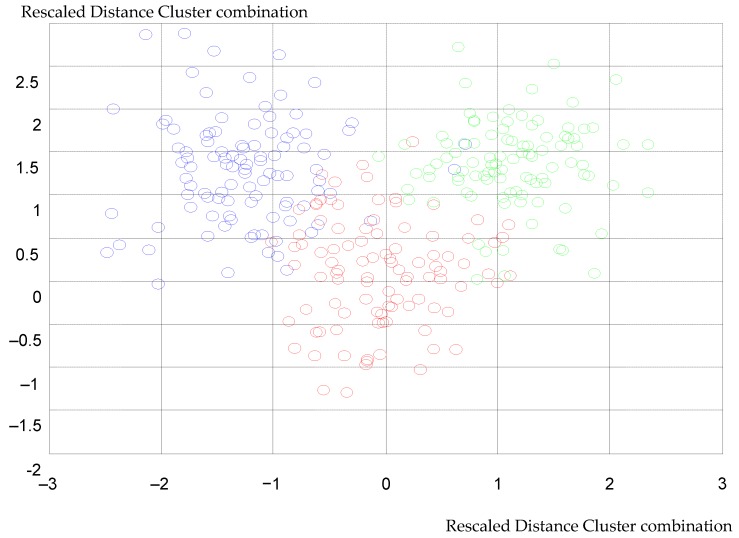
Improved KNN (k-Nearest Neighbor) clustering data of vehicle speed.

**Figure 6 ijerph-16-03647-f006:**
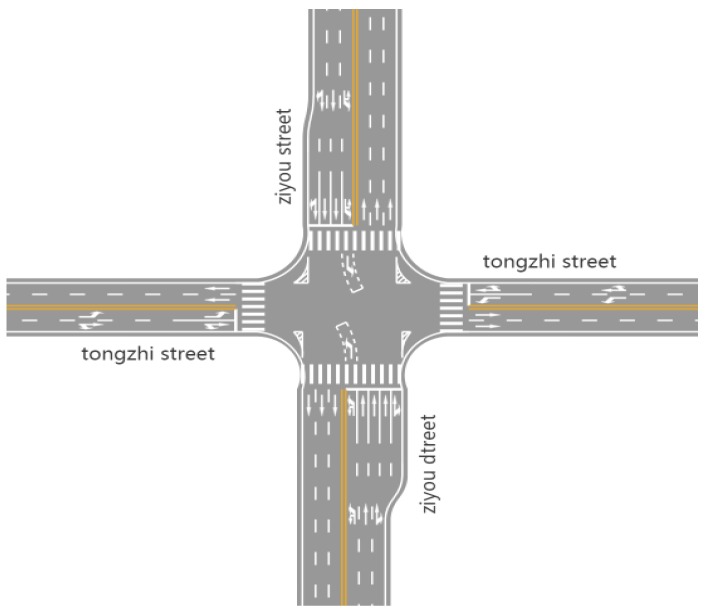
Sample diagram of intersection.

**Figure 7 ijerph-16-03647-f007:**
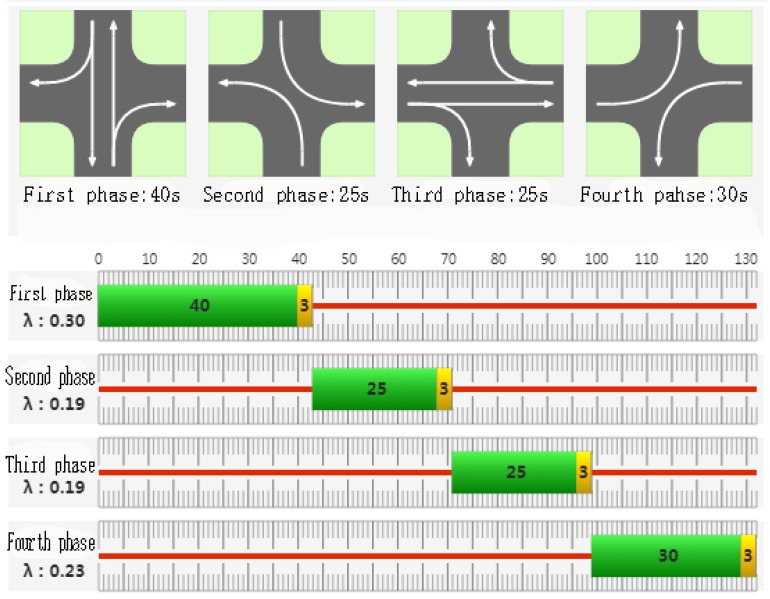
The actual signal timing diagram of the test intersection.

**Figure 8 ijerph-16-03647-f008:**
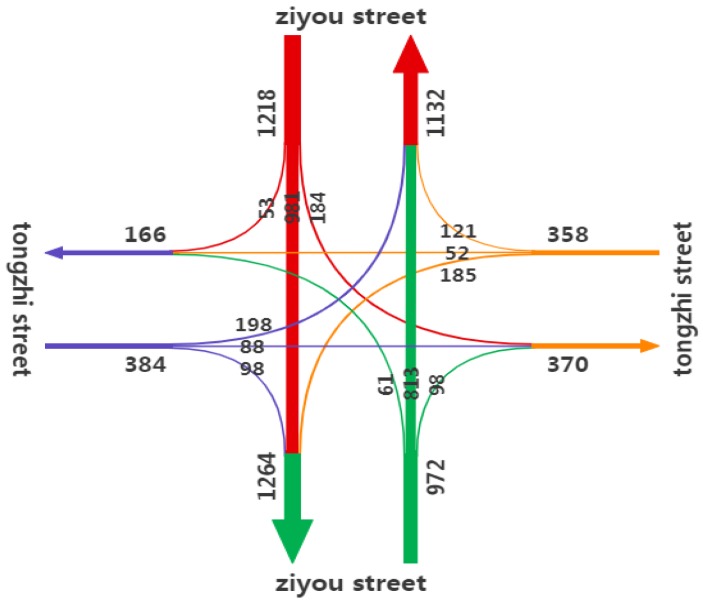
Test intersection traffic flow.

**Table 1 ijerph-16-03647-t001:** Description of GPS data of experimental vehicles from February 14 to February 15.

Date	ID	Longitude	Latitude	Velocity	Directional Angle
20190214	578044	125.316107	43.6884	45	0
20190214	563486	125.422085	43.903562	46	90
20190214	403211	125.654715	43.288522	0	114
20190214	214013	125.84268	44.657383	0	0

**Table 2 ijerph-16-03647-t002:** Exhaust emission rate of various types of cars under different specific powers.

Specific Power Interval (kw/t)	Emissions (g/s)			
	CO	CO_2_	NO_x_	HC
(−∞,−2)	0.0110	1.5437	0.0010	0.0009
[−2,0)	0.0087	1.6044	0.0010	0.0009
[0,1)	0.004 7	1.1308	0.0004	0.0008
[1,4)	0.0122	2.3863	0.0016	0.0010
[4,7)	0.0167	3.2102	0.0026	0.0013
[7,10)	0.0233	3.9577	0.0038	0.0017
[10,13)	0.0293	4.7520	0.0051	0.0021
[13,16)	0.0369	5.3742	0.0064	0.0023
[16,19)	0.0495	5.9400	0.0077	0.0028
[19,23)	0.0638	6.4275	0.0099	0.0030
[23,28)	0.1054	7.0660	0.0127	0.0038
[28,33)	0.2478	7.6177	0.0144	0.0046
[33,39)	0.4131	8.3224	0.0156	0.0057
[39,+∞)	0.6247	8.4750	0.0167	0.0072

**Table 3 ijerph-16-03647-t003:** Four models corresponding to average specific power.

Vehicle Name	Volkswagen Magotan	FAW Jetta	FAW Pentium	Toyota Corolla
Specific Power	76.4	58.9	88.7	90

**Table 4 ijerph-16-03647-t004:** Related specific power exhaust emissions.

Emission Gas Type	CO	CO_2_	NO_x_	HC
g/s	0.6247	8.4750	0.0167	0.0072

**Table 5 ijerph-16-03647-t005:** Total emissions under actual vehicle speed.

CO_2_ Emission (g)	NO_x_ Emission (g)
15,4024	1459

**Table 6 ijerph-16-03647-t006:** The exhaust emissions obtained by the three methods.

	Actual Measured Value	Improved Value of MOVE Emission Model Based on GPS Data	MOVE Emission Model Value
CO_2_ Emission (g)	154,024	148,112	144,534
NO_x_ Emission (g)	1459	1386	1326

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
