# Peer review of "Estimation of Emissions at Signalized Intersections Using an Improved MOVES Model with GPS Data"

_ijerph, 2019, doi:10.3390/ijerph16193647_

Round 1

Reviewer 1 Report

This paper estimates vehicle fuel consumption and emissions using GPS traffic data. The author tries to improve traditional MOVE model by incorporating GPS data into the model . Vehicle speed information is added to VSP model to improve the road section and vehicle information in the MOVE model. However, it’s a little difficult to follow the author’s discussion in Section  2.3.2.

 Can the author elaborate on why the process of driving to a signal-controlled intersection can be divided into two cases?  I didn’t see a clear explanation of the breakdown of  networked vehicles into four conditions (line 190) either?

This part is the core of the paper’s argument and contribution. So, I believe the author should spend more effort on this part of the discussion and make sure we understand the contribution of this research more clearly. 

Some minor points about the formatting and figures: 

What do the axes of figure 2 & figure 3 represent? I cannot find an explanation in the manuscript. English of this manuscript should be improved.There are  typos and formatting errors in the manuscript: for example, line 161 ,line 257, line 226, line 300

Author Response

Dear Reviewer:

Manuscript ID: ijerph-587657

Title: “Estimation of Emissions at Signalized Intersections Using an Improved MOVES model with GPS Data”

We wish to express our very deep appreciation, and the appreciation of all of us, to your great efforts and suggestions for our manuscript. They are valuable and very helpful for revising and improving our paper, as well as the important guiding to our researches.

The following is a point-to-point response to your comments and responses are in red. The modification marked in red in revised version.

Response to comment: (.The process of driving to a signal-controlled intersection can be divided into two cases)

Response: We are very sorry for our negligence of explanation. Nevertheless, every vehicle speed can be quickly classified from the data. But the macro vehicle queue does not reflect precise emissions. Different specific power interval was corresponding to different speed section from Table 2. Vehicles wouldn’t cause idle state when vehicles pass through the intersection in the green light time. In this state, vehicle queue running state is relatively simple. Using vehicle traffic speed and time could analyze vehicle queue’s exhaust emission. But vehicles in red time couldn’t pass through the intersection. The vehicle status in the vehicle queue was correlated with each other. The macro method was used to analyze vehicle queue and then the state of each car was analyzed from the four micro states to combine with VSP model. On the whole, the process of driving to a signal-controlled intersection could be divided into the following two cases. See line 200 to line 209.

Response to comment: (Don’t see a clear explanation of the breakdown of networked vehicles into four conditions (line 190))

Response: Macroscopically analyzes the number of vehicles idling during red light time. The number of vehicles idling at signalized intersections can be quickly statistics by using signal timing.

Response to comment: (What do the axes of figure 2 & figure 3 represent?)

Response: We are very sorry for our negligence of axis’s explanation. Two figures are added to illustrate intersection contour and the algorithm process, so Figure. 2 becomes figure 4 and figure. 3 becomes figure. 5. The figure 4’s horizontal axis represents the speed of the vehicle and the vertical axis represents distance between sample points. The figure 5’s horizontal and vertical axis represents rescaled distance cluster combine. See line 141 to line 146.

Response to comment: (There are typos and formatting errors in the manuscript: for example, line 161 ,line 257, line 226, line 300)

Response: We have corrected errors about format and grammar, The modification marked in red.

We tried our best to improve the manuscript and made some changes in the manuscript. These changes will not influence the content and framework of the paper. And here we did not list the changes but marked in red in revised paper. We appreciate for your warm work earnestly, and hope that the correction will meet with approval. Thank you for your time and patience. I look forward to receiving your letter.

Once again, we would like to thank you for the constructive comments and suggestions. Please feel free to contact us with any questions. We are looking forward to your reply.

Yours sincerely,

Authors

Reviewer 2 Report

This manuscript is very interesting and has the potential to reach a wide audience. 

In this manuscript, it was used GPS from the Internet company Didi which offers services of passenger transportation, very popular in China. The data, which has a frequency of 10 s,  was processed at an intersection in the city of Changchun (Jilin Province, China) and then the vehicular emissions were estimated using the model MOtor Vehicle Emissions Simulator (MOVES). 

I have the following concerns and comments:

The name of the model is MOVES, not MOVE (https://www.epa.gov/moves/latest-version-motor-vehicle-emission-simulator-moves and add the respective citation. The frequency of the observations is said to have an approximate sampling of 10 seconds (line 70), however, the VSP methodology was designed to work with GPS recordings of 1 second. I know that obtaining that with this frequency from Internet services might be difficult, nevertheless, the authors should add a statistical description of the frequency of the data. Figure 1 does not have the source. Also, indicate in the map the location of the intersection studied. This might be done with 1 figure with two panels. First panel the map of the city and second a zoom of the study intersections. This might be helpful for readers outside of China. It is mention that Didi is an Internet company without mentioning that consists in passenger cars transportation. Please, mention this in the text. Table 1 has no title. Add it and add a column indicating the period of time of the recordins. Figure 2 says the vehicle speed in m/s, with many observations of 100 m/s, which results in 360 km/h. Fix typo. There is no citation for equation 3. Table 2 would look better in a Figure. Improve English in the whole text Another paper that might be included in the reference is the model VEIN which has emission factors for China: Wu, L., Chang, M., Wang, X., Hang, J., and Zhang, J.: Development of a real-time on-road emission (ROE v1.0)model for street-scale air quality modeling based on dynamic traffic big data, Geosci. Model Dev. Discuss., https://doi.org/10.5194/gmd-2019-74, in review, 2019

Author Response

Dear Reviewer:

Manuscript ID: ijerph-587657

Title: “Estimation of Emissions at Signalized Intersections Using an Improved MOVES model with GPS Data”

We wish to express our very deep appreciation, and the appreciation of all of us, to your great efforts and suggestions for our manuscript. They are valuable and very helpful for revising and improving our paper, as well as the important guiding to our researches.

The following is a point-to-point response to your comments and responses are in red. The modification marked in red in revised version.

Response to comment: (.The name of the model is MOVES, not MOVE add the respective citation)

Response: We are very sorry for this error. We have modified MOVE to MOVES and add the respective citation. See the Refs. [15-16].

Response to comment: (.The frequency of the observations is said to have an approximate sampling of 10 seconds (line 70),)

Response: We are very sorry for our negligence of explanation. Different time intervals are selected for different road conditions. The actual return transmission interval is set at about 1s to ensure the accuracy of the intersection emission test. The interval is not set at 1s on the normal driving road. In fact, considering that the shorter the time needs, the greater the power of the equipment and the more prone to failure of the equipment operation. Subsequent research uses 1s as the interval for test intersection validation. See line 86 to line 93.

Response to comment: ( Figure 1 does not have the source. Also, indicate in the map the location of the intersection studied. This might be done with 1 figure with two panels. First panel the map of the city and second a zoom of the study intersections)

Response: It is really true as reviewer suggested. Figure.1 is the distribution map of the intersections from Google map. The Studied intersection is Ziyou road and Tongzhi Street. Figure 2 is an aerial map of the study intersection. We have added test intersections as figure 2.See line 80 to line 85.

Response to comment: (It is mention that Didi is an Internet company without mentioning that consists in passenger cars transportation.)

Response: We indicate Didi is a technology company focused on transportation. And Didi establishes a flexible travel mode of users by the internet. See line 73 to line 74.

Response to comment: (Table 1 has no title. Add it and add a column indicating the period of time of the recording)

Response: We have made correction according to the Reviewer's comments. We indicate test data records the evening rush hour from 5pm to 6 pm. See line 75,line 102.

Response to comment: (Figure 2 says the vehicle speed in m/s, with many observations of 100 m/s, which results in 360 km/h..)

Response: We are very sorry for our incorrect writing. The figure’s horizontal axis represents the speed of the vehicle and the vertical axis represents distance between sample points. The unit of velocity is kilometers per hour in figures. See line 141 to line 146.

Response to comment: (There is no citation for equation 3.)

Response: We are very sorry for our negligence. We have added the respective citation. See line 164.

Response to comment: (Another paper that might be included in the reference)

Response: We have added references according to the reviewer 's comments.

ZHAO Q, CHEN Q, WANG L. Real-Time Prediction of Fuel Consumption Based on Digital Map API [J]. Applied Sciences, 2019, 9(7):

KANARACHOS S, CHRISTOPOULOS S-R G, CHRONEOS A. Smartphones as an integrated platform for monitoring driver behaviour: The role of sensor fusion and connectivity [J]. Transportation Research Part C: Emerging Technologies, 2018, 95(867-82.

KANARACHOS S, MATHEW J, FITZPATRICK M E. Instantaneous vehicle fuel consumption estimation using smartphones and recurrent neural networks [J]. Expert Systems with Applications, 2019, 120(436-47.

PING P, QIN W, XU Y, et al. Impact of Driver Behavior on Fuel Consumption: Classification, Evaluation and Prediction Using Machine Learning [J]. IEEE Access, 2019, 7(78515-32.

ZHENG F, LI J, VAN ZUYLEN H, et al. Influence of driver characteristics on emissions and fuel consumption [J]. Transportation Research Procedia, 2017, 27(624-31.

WU L, CHANG M, WANG X, et al. Development of a real-time on-road emission (ROE v1.0)model for street-scale air quality modeling based on dynamic traffic big data [J]. Geoscientific Model Development Discussions, 2019, 1-19.

Response to comment: (There are typos and formatting errors in the manuscript)

Response: We have corrected errors about format and grammar, The modification marked in red in revised version.

We tried our best to improve the manuscript and made some changes in the manuscript. These changes will not influence the content and framework of the paper. And here we did not list the changes but marked in red in revised paper. We appreciate for your warm work earnestly, and hope that the correction will meet with approval. Thank you for your time and patience. I look forward to receiving your letter.

Once again, we would like to thank you for the constructive comments and suggestions. Please feel free to contact us with any questions. We are looking forward to your reply.

Yours sincerely,

Authors

Reviewer 3 Report

First of all I would like to thank the authors for their time and efforts to prepare this manuscript. The topic is very interesting and timely but needs to be revised in order to meet the quality standards. Some of my suggestions include:

·       Overall: Careful proofreading of the manuscript by a native English speaker, if possible. There are several

·       Section 1: The literature review needs to include key recent contributions and relate it to it, e.g.:

https://www.sciencedirect.com/science/article/pii/S0957417418307681

https://ieeexplore.ieee.org/document/8727915

https://www.mdpi.com/2076-3417/9/7/1369

https://www.sciencedirect.com/science/article/pii/S0968090X18303954

·       Section 2.1 The authors need to explain in greater detail the experimental plan: how many vehicles, what type of vehicles, which sensors, for which period, e.t.c.. The authors need to also summarise the range of measurement they did e.g. speed range, e.t.c.

·       Section 2.2: The algorithm needs to be described in the form of a pseudocode rather than bullet points. It is not clear what the meaning of all symbols used is. The authors need to denote the variables.

·       Figure 2 and 3. It is not clear what the y axis represents

·       Table 2: It is not clear where this information comes from. The authors need to refer to their source. Furthermore, aren’t emissions also a function of temperature like the authors indicated in the beginning?

·       Section 2.3.2: If the sampling interval is 10 s how is it possible to have accurate measurements of deceleration and acceleration? Is it possible for the authors to discuss this aspect.

·       Equations (14), (15), (17) and (18) look awkward. I can’t find Equation (16)

·        Line 240: MOBILE

·       Section 3.1: Paragraph format is wrong

·       Please revise Figures so that fonts appear in high quality

·       The model checking section is unclear. The authors need to add clarity and improve the description.

·       The discussion section needs to be improved. In particular, a systematic comparison between the standard MOVE model and the proposed one for different driving conditions and in general needs to be made

Author Response

Dear Reviewer:

Manuscript ID: ijerph-587657

Title: “Estimation of Emissions at Signalized Intersections Using an Improved MOVES model with GPS Data”

We wish to express our very deep appreciation, and the appreciation of all of us, to your great efforts and suggestions for our manuscript. They are valuable and very helpful for revising and improving our paper, as well as the important guiding to our researches.

The following is a point-to-point response to your comments and responses are in red. The modification marked in red in revised version.

Response to comment: (Section 1: The literature review needs to include key recent contributions and relate it to it,)

Response: We have added references according to the reviewer 's comments .We have added some recent key contributions references

ZHAO Q, CHEN Q, WANG L. Real-Time Prediction of Fuel Consumption Based on Digital Map API [J]. Applied Sciences, 2019, 9(7):

KANARACHOS S, CHRISTOPOULOS S-R G, CHRONEOS A. Smartphones as an integrated platform for monitoring driver behaviour: The role of sensor fusion and connectivity [J]. Transportation Research Part C: Emerging Technologies, 2018, 95(867-82.

KANARACHOS S, MATHEW J, FITZPATRICK M E. Instantaneous vehicle fuel consumption estimation using smartphones and recurrent neural networks [J]. Expert Systems with Applications, 2019, 120(436-47.

PING P, QIN W, XU Y, et al. Impact of Driver Behavior on Fuel Consumption: Classification, Evaluation and Prediction Using Machine Learning [J]. IEEE Access, 2019, 7(78515-32.

ZHENG F, LI J, VAN ZUYLEN H, et al. Influence of driver characteristics on emissions and fuel consumption [J]. Transportation Research Procedia, 2017, 27(624-31.

WU L, CHANG M, WANG X, et al. Development of a real-time on-road emission (ROE v1.0)model for street-scale air quality modeling based on dynamic traffic big data [J]. Geoscientific Model Development Discussions, 2019, 1-19.

WU Y, SONG G, YU L. Sensitive analysis of emission rates in MOVES for developing site-specific emission database [J]. Transportation Research Part D: Transport and Environment, 2014, 32(193-206).

XU J, SALEH M, WANG A, et al. Embedding local driving behaviour in regional emission models to increase the robustness of on-road emission inventories [J]. Transportation Research Part D: Transport and Environment, 2019, 73(1-14).

Response to comment: (Section 2.1 The authors need to explain in greater detail the experimental plan: how many vehicles, what type of vehicles, which sensors, for which period, e.t.c.. The authors need to also summarise the range of measurement they did e.g. speed range, e.t.c.)

Response: We have made correction according to the reviewer's comments. The data comes from the GPS data of the vehicles in Changchun collected by Didi Internet Company. Didi is a technology company focused on transportation. It is worth mentioning that Didi establishes a flexible travel mode of users by the internet. The vehicle data of the network records the driving data of multiple vehicles in one day. Test data records the evening rush hour from 5pm to 6 pm. The GPS data contains 86 intersections. The types of vehicle have Volkswagen Magotan, FAW Jetta, FAW Pentium and Toyota Corolla. The actual test data included 562 Volkswagen Magotan vehicles, 1335 FAW Jetta vehicles, 723 FAW Pentium vehicles, and 289 Toyota Corolla vehicles. The speed range of the vehicles is less than 100km/h. The test drivers of the vehicles were in normal driving state. Figure.1 is the distribution map of the intersections from Google map. The Studied intersection is Ziyou road and Tongzhi Street. Figure 2 is an aerial map of the study intersection.The actual vehicle emission testing instrument adopted in this study is the OEM-2100 produced by CATI company in the United States. The instrument calculates the exhaust volume flow rate according to the measured engine parameter data, and calculates the instantaneous mass emission rate based on the measured exhaust volume percentage.We have added statements in Section 2.1

Response to comment: (Section 2.2: The algorithm needs to be described in the form of a pseudocode rather than bullet points. It is not clear what the meaning of all symbols used is. The authors need to denote the variables.)

Response: We have added flow chart as figure3 according to the reviewer's comments .We have the horizontal and vertical coordinates of the image are explained. See section 2.3.2.

Response to comment: ( Figure 2 and 3. It is not clear what the y axis represents.)

Response: Two figures are added to illustrate intersection contour and the algorithm process, so Figure. 2 becomes figure 4 and figure. 3 becomes figure. 5. The figure 4’s horizontal axis represents the speed of the vehicle and the vertical axis represents distance between sample points. The figure 5’s horizontal and vertical axis represents rescaled distance cluster combine. See line 141 to line 146.

Response to comment: ( Table 2: It is not clear where this information comes from.)

Response: We have added citation about Table 2.See line 177.

Response to comment: (Aren’t emissions also a function of temperature like the authors indicated in the beginning.)

Response: Although different temperatures could have little impact on emission, the current car fuel tank has temperature protection and the exhaust emission temperature is higher than temperature. The GPS data is used in simulation without separated analysis. See line 192 to line 195.

Response to comment: ( Section 2.3.2: If the sampling interval is 10 s how is it possible to have accurate measurements of deceleration and acceleration? Is it possible for the authors to discuss this aspect.)

Response: We are very sorry for our negligence of explanation. Different time intervals are selected for different road conditions. The actual return transmission interval is set at about 1s to ensure the accuracy of the intersection emission test. The interval is not set at 1s on the normal driving road. In fact, considering that the shorter the time needs, the greater the power of the equipment and the more prone to failure of the equipment operation. Subsequent research uses 1s as the interval for test intersection validation. See line 86 to line 93.

Response to comment: ( Equations (14), (15), (17) and (18) look awkward. I can’t find Equation (16))

Response: We are very sorry for equations’ numbering.  Equations (14), (15), (17) and (18) mean different traffic parameter to constitute different vehicles conditions emission.

Response to comment: (Line 240: MOBILE)

Response:We have corrected error. MOBILE have corrected to MOVES. See line 283.

Response to comment: (Section 3.1: Paragraph format is wrong. Please revise Figures so that fonts appear in high quality)

Response: We have corrected format. See line 275.

Response to comment: ( The model checking section is unclear. The authors need to add clarity and improve the description.)

Response: We have improved the description about checking section. See line 309 to line 312.

Response to comment: ( The discussion section needs to be improved. In particular, a systematic comparison between the standard MOVE model and the proposed one for different driving conditions and in general needs to be made)

Response: In this paper, through joining real-time GPS data analyzes the vehicles entering the intersection emissions. .Compared with the traditional MOVES model, the traditional model studies vehicle emissions by using variables such as vehicle model environmental, speed, etc. Moves model does not consider the change of vehicle conditions in the changing process of signalized intersections and the necessary delay caused by the signal control at the intersection. The real-time GPS data added by the improved model can effectively improve the accuracy of emission quantification, and analyze emissions by combining the timing of different signal intersections. Combined with the process of vehicles entering the intersection, it can reasonably analyze the emission of vehicles under different conditions .The exhaust emission obtained by the three methods can be obtained. The error between the emission calculated based on the GPS improved model and the actual measured value is significantly smaller than that obtained by the traditional MOVES model. This further verifies that the GPS-based improved model is better. Adding real-time GPS data to the model can improve the accuracy of exhaust emission and related parameters in macroscopic and microscopic level. It’s significant to improve the efficiency of GPS data processing, which is an important direction in the future. See line 338 to line 352.

We tried our best to improve the manuscript and made some changes in the manuscript. These changes will not influence the content and framework of the paper. And here we did not list the changes but marked in red in revised paper. We appreciate for your warm work earnestly, and hope that the correction will meet with approval. Thank you for your time and patience. I look forward to receiving your letter.

Once again, we would like to thank you for the constructive comments and suggestions. Please feel free to contact us with any questions. We are looking forward to your reply.

Yours sincerely,

Authors

Round 2

Reviewer 1 Report

I have no further comments about this manuscript.